# Intelligent Pick-and-Place System Using MobileNet

**Fan Hong** [1,*]**, Donavan Wei Liang Tay** [2] **and Alfred Ang** [3]

1    James Watt School of Engineering, University of Glasgow, Glasgow G12 8QQ, Scotland, UK
2    Engineering Cluster, Singapore Institute of Technology, Woodlands 737715, Singapore
3    Tertiary Infotech Pte. Ltd., Woodlands 737715, Singapore
\*    Correspondence: fan.hong@glasgow.ac.uk; Tel.: +65-69086035

**Abstract:** The current development of a robotic arm solution for the manufacturing industry requires performing pick-and-place operations for work pieces varying in size, shape, and color across different stages of manufacturing processes. It aims to reduce or eliminate the human error and human intervention in order to save manpower costs and enhance safety at the workplace. Machine learning has become more and more prominent for object recognition in these pick-and-place applications with the aid of imaging devices and advances in the image processing hardware. One of the key tasks in object recognition is feature extraction and object classification based on convolutional neural network (CNN) models, which are generally computationally intensive. In this paper, an intelligent object detection and picking system based on MobileNet is developed and integrated into an educational six-axis robotic arm, which requires less computation resources. An experimental test is conducted on six-axis robotic arm called Niryo One to train the model and identify three objects with difference shapes and colors. It is shown by the confusion matrix that the MobileNet model achieves an accuracy of 91%, a dramatic improvement compared to 65% of the Niryo One's original sequential model. The statistical study also shows the MobileNet can achieve a higher precision with more clustered spread of accuracy.

**Keywords:** machine learning; MobileNet; robotic arm; pick-and-place system

## 1. Introduction

In modern manufacturing industries, pick-and-place robots are commonly used due to their capabilities of automation and expedition of the process of picking and placing items in desired locations [1]. In addition, using pick-and-place robots to replace humans is beneficial especially in arduous, highly repetitive, and hazardous tasks, in order to reduce the labor costs and ensure consistency in the quality control of tasks.

There are different types of pick-and-place robots such as cartesian robots [2], fast pick-and-place robots [3], robotic arm, delta (or parallel) robots [4], and collaborative robots (cobot) [5]. Pick-and-place robots have several dedicated parts including the robot arm tool, end effector, actuators, sensors, and controllers [6].

In manufacturing, various items are often transported via a conveyor belt before entering into the robots' work envelope. These items or work pieces vary in size, shape, color across different stages of manufacturing processes. Multiple automated pick-and-place robots are commonly mounted on a stand so that they can reach the entire aera of operation to perform the required tasks. Robots are usually equipped with advanced vision systems that enable them to identify different objects through object detection and recognition training. This capability allows robots to identify different items and parts accurately based on the orientation, location, color, and size. In addition, different end-effectors can be designed depending on their respective applications in order to segregate the objects to their respective target locations.

As technology advances, machine learning has become more prominent around the world. Cameras have been used for object detection so that robots will be able to operate au-

tonomously based on information gathered from cameras [7]. This is known as "computer vision" and has attraction a great amount of attention in academia and industries. With the advances in hardware, such as Graphics Processing Units (GPUs), which allow highly parallel computations in machine learning to be accelerated greatly, computer vision also starts to leverage on advances in machine learning and greatly increases the capabilities and possibilities of what it can achieve.

Object recognition is actually a collection of computer vision tasks to identify objects from images. These tasks can be categorized generally as feature extraction and object classification [8]. The feature extraction stage is composed of a filter bank, a non-linear operation, and some feature pooling operations, together forming the typical convolutional neural networks (CNN). The output of the pooling operation is classified by either supervised or unsupervised learning classifiers, which are under the object classification stage [9].

Convolutional neural networks (CNN) have achieved many successful applications, especially in imagine recognition, such as the AlexNet model [10], which has around sixty million parameters and eight layers; the VGG model [11] with sixteen layers; as well as GoogleNet [12] and ResNet [13]. However, the requirements for high accuracy and powerful feature extractions have made CNN algorithms computationally intensive with substantial memory footprints [14]. It is inefficient to implement these algorithms on embedded platforms such as robotics, unmanned vehicles, and IoT devices, which are subject to constraints in power consumption, memory, resources, and timing. MobileNet, proposed by Google, is a class of efficient and lightweight neural network model dedicated to mobile and embedded vision applications. It has considerably reduced the number of feature parameters and required memory footprint based on streamlined architecture that uses depth-wise separable convolutions to build neural networks [16].

Motivated by previous works on machine learning using different training models, an intelligent object detection and picking system based on MobileNet will be developed and implemented on an educational six-axis robotic arm called Niryo One [17]. The main contributions of the paper lie in the following:

(i) The MobileNet model for training purposes is built with a properly chosen number of neurons and activation functions;

(ii) The training program using MobileNet model is coded in Python in the Tensor-Flow platform;

(iii) The training program for object detection is integrated in the existing Niryo One robotic system to perform pick-and-place applications;

(iv) The experimental results show that the MobileNet model achieves an accuracy of 91%, a dramatic improvement compared with 65% of the robot system's existing sequential model;

(v) The statistical study shows the MobileNet can achieve a higher precision with a more clustered spread of accuracy.

The rest of the paper is organized as follows. The problem formulation and methodology are given in Section 2. The code structure of the developed Python program is presented in Section 3. The experimental result is shown in Section 4, followed by Section 5 with the discussion of the problems, challenges, and future improvements of the work. Lastly, Section 6 concludes the paper.

## 2. Problem Formulation and Methodology

The Niryo One uses TensorFlow, an open-source machine learning tool developed by Google, to recognize multiple objects on its workspace. It makes use of its vision set, artificial intelligence, image processing, and machine learning [18]. In its original program, images of the objects to be identified will go through some pre-processing before being fed into a Python program called "training.py" to train the neural network to obtain a TensorFlow model.

The model is a sequential class model that consists of a sequence of layers, one after the other. It has a certain number of inputs, a hidden layer with certain number of neurons, and an output layer with a single neuron. Additional layers can be created and added to the model.

In this paper, the MobileNet model is built and coded in Python in the TensorFlow platform, as well as Keras library and APIs. The MobileNet model will be trained using image data of the objects and the trained model will be then be used to recognize the different objects on the workspace.

Figure 1 shows the different layers in the convolutional neural network architecture. The first layer is the input layer, which feeds images to the second layer, i.e., the convolution layer. The third layer is the flatten layer, the fourth is fully connected layer, the fifth is the dropout layer, and the last layer is the output layer. The first, second, and third layers perform image extraction, and the fourth, fifth, and sixth layers perform image classification.

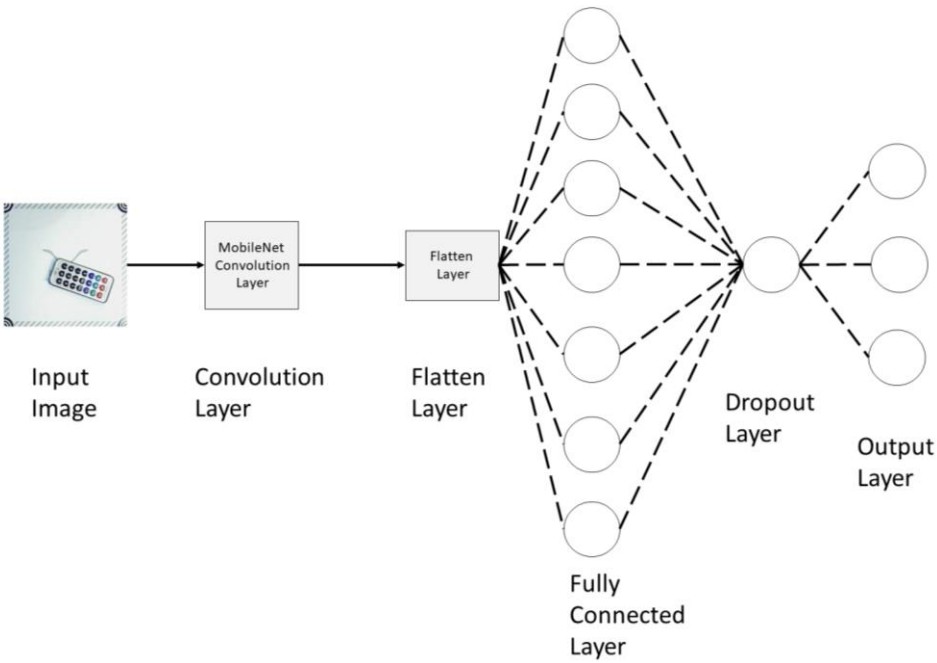

**Figure 1.** Convolutional neural network architecture for the MobileNet model [15].

The function of each layer is described as follows.

The MobileNet model is a neural network model that uses a new kind of convolution layer, known as depth-wise separable convolution, as its basic unit. Its depth-wise separable convolution has two layers: depth-wise convolution and point convolution. The depth-wise convolution filter performs a single convolution on each input channel, and the point convolution filter combines the output of the depth-wise convolution linearly with $1 * 1$ convolutions [16].

Convolution, a form of linear operation used for feature extraction, uses a small array of numbers known as kernels, applied across the input image. This array of numbers is called a tensor. A feature map can then be calculated using the input tensor and the kernel [19]. A simple example of using a dot product between the input matrix and the kernel to generate the output feature map was shown in [20] (Figure 6.6). The feature map is then passed to the third layer.

The third layer or the flatten layer converts the feature map into a single one-dimensional layer.

The fourth layer is the fully connected layer. The output of the flatten layer is passed into numerous amounts of neurons. In [21], 128 neurons with a rectified activation function

or rectified linear unit (ReLU) were used. The equation for a single neuron activation using "ReLU" is given by

$$h_\theta(x) = \text{ReLU}(w^T x + b), \text{ where } w \in \mathbb{R}^d, \ b \in \mathbb{R}, \text{ and } \theta = (w, b) \tag{1}$$

where "$b$" is the bias and "$w$" is the weight.

The output of the fourth layer is fed into the fifth layer. The fifth layer is the dropout layer, and it is used to prevent overfitting. According to [22], it has been proven that a dropout rate of 0.5 is ideal for large neural networks.

The output of the dropout layer is fed into the input of the last layer, the output layer. Three neurons are used in this layer, as the model is used to predict three types of items. The output layer uses the "SoftMax" activation function, which is given by

$$P(y = j | \Theta^{(i)}) = \frac{e^{\Theta^{(i)}}}{\sum_{j=0}^{k} e^{\Theta_k^{(i)}}} \tag{2}$$

where $\Theta = w^T x$ is the sum of scores and "$i$" is the input parameter [23]. According to [24], SoftMax activation function outputs decimal probabilities to each class of labels and the sum of all probabilities adds up to one. This allows for the model to view an item and produce probabilities of what the image represents.

The confusion matrix is one of the major metrics to determine the performance of the classification process in CNN [25]. The confusion matrix uses true positives, false positives, true negatives, and false negatives to determine the precision, recall, accuracy, and f1-score. True positive means the CNN model correctly predicts the object's name while false positive means the model falsely predicts the object's name. True negative means that the model predicts a negatively result and the test says the model is true, while false negative means that the model predicts a negative result, but the test says the model is false [26].

The number of actual occurrences in the dataset is called support. Precision is the ratio between the true positive result and the total positive guesses, and is calculated as below

$$\text{Precision} = \frac{\text{True Positive}}{\text{True Positive} + \text{False Positive}}$$

Recall is the ratio between the true positive result and the summation of true positive with false negatives, which is calculated as

$$\text{Recall} = \frac{\text{True Positive}}{\text{True Positive} + \text{False Negative}}$$

The f1-score is known as harmonic mean and is used to measure the mean score between precision and recall. It is calculated as

$$\text{f1} - \text{score} = \frac{2 \times (\text{Precision} \times \text{Recall})}{\text{Precision} + \text{Recall}}$$

Accuracy is the percentage of how many accurate predictions the CNN model has given over the total number of test cases

$$\text{Accuracy} = \frac{\text{True Positive}}{\text{True Positive} + \text{True Negative} + \text{False Positive} + \text{False Negative}}$$

## 3. Program and Code Structure

In this paper, the original Python program called "training.py" in Niryo One will be replaced by a new program called "MobileNet_Training.py" to train the image data using MobileNet. A detailed explanation of the code structure and functionalities are given as follows.

Firstly, Figure 2 shows the flow chart for the machine learning function called "MobileNet_Train()".

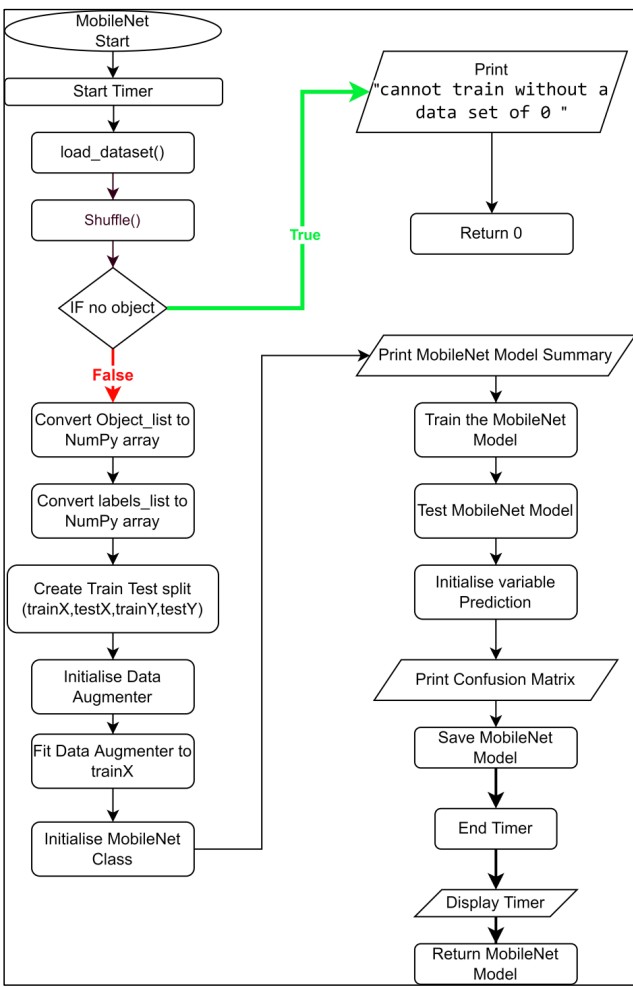

**Figure 2.** Function "MobileNet_Train()" in "MobileNet_Training.py".

As shown in Figure 2, the function "MobileNet_Train()" performs the following functionalities:

1.  Initialize variable "start_time" and pass current time to it.
2.  Call function "load_dataset()" and pass the return values to the variables "data_set" and "objects_names".
3.  Call the shuffle function and pass the return values to variables "objects_list", "labels_list", and "files_names".
4.  If there are no objects in "objects_list", print "cannot train without a data set of 0".
5.  Otherwise, convert the "objects_list" and "labels_list" to NumPy arrays.
6.  Create a Train Test Split and pass the return values to variables "trainX", "testX", "trainY" and "testY".
7.  Call function "ImageDataGenerator()" and pass it to a variable called "datagen", which creates a function to augment data.
8.  Fit the data augmenter "datagen" to the training data "trainX", which augments the training data and creates more training data.
9.  Create a variable called "model" and pass an instance of the MobileNet model into it.
10. Print the model summary.
11. Train the MobileNet Model by calling the function model.fit().
12. Create a variable "prediction", which is used to predict all of the test images "testX" and return the highest predicted value for each image.

13. Print the confusion matrix by calling the function "classification_report()".
14. Save the MobileNet Model.
15. Create a variable "end_time" and pass the difference between the current time and the "start_time".
16. Print "end_time".
17. Return the MobileNet model.

Figure 3 shows the flow chat of a function called "load_dataset()" in "MobileNet_Training.py". This function is from Niryo One's original code [27], and is used to iterate through all the folders in the "data" folder and extract all the class names and perform object extraction of all the images in each folder.

1. Create a list called "objects_names" and add all the names of the folder in the list.
2. Prints all the names in the "objects_names" list.
3. Create "objects_list", "labels_list" and "files_names", which are used to store object images, object labels, and object names respectively.
4. Try to make a data mask directory. If it already exists, go to the next line of code.
5. Iterate over all the names in the "object_names" list:

   a. Create a variable called "list_dir", which stores the address of the folder where the images are stored.
   b. Print the name of the folder and the number of images in the folder.
   c. Try to make a "data_mask/object name" directory. If it exists, go to the next line of code.
   d. Iterate through all the images in "list_dir":

      i. Create a variable called "img" which is used to read the image.
      ii. Call "utils.standardize_img()". This function originates from Niryo One's library and is used to normalize the color of "img".
      iii. Call "utils.objs_mask()" and pass the image "img" to it. This function is from Niryo One's library and is used to create extract regions of interest from images. The returned result is passed to the variable "mask".
      iv. Call "utils.extract_objs()" and pass "img" and "mask" to it. This function comes from Niryo One's library and is used to find shapes in an image. If an object is found from an image, the code will create a rectangle around the object, make the image in a vertical orientation and return the image. The returned image is passed to the variable "objs".
      v. Iterate through all pixel in the image "objs":

         (1) Save image "img" in folder "data_mask/object Name/number/ file_name".
         (2) Resize the image to (64,64) pixel.
         (3) Create a NumPy array called "img_float" full of zeros with arguments (64, 64, 3), np.float32. This indicates that the pixel size is 64 by 64, and the number of colour channels is 3, i.e., Red, Green, and Blue in this case.
         (4) Scale the colour of the image "img" between 0 and 1 and pass it to the NumPy array "img_float".
         (5) Create a NumPy array called "label" full of zeros with the size of "object_names".
         (6) Set label[obj_id] to 1, which changes the label names to binary form. Each index will represent a class name.
         (7) Add the NumPy array "img_float" which contains the images to "object_list".
         (8) Add the NumPy array "label" which contains the label index to "label_list".
         (9) Add the total number of images and file name to the list "files_names".

       vi.      Print the number of objects detected, which can inform the user how many objects that function "utils.extract_objs()" has detected.

       vii.     Print "|", which informs the user that the algorithm is running the next loop.

    e.      Print "" this creates a new line. This is to indicate that the algorithm is performing object extraction for the next class name.

    f.      Increment the variable "obj_id", which is used to indicate the index of the labels.

   6.     Return "objects_list", "labels_list", files_names", and "objects_names".

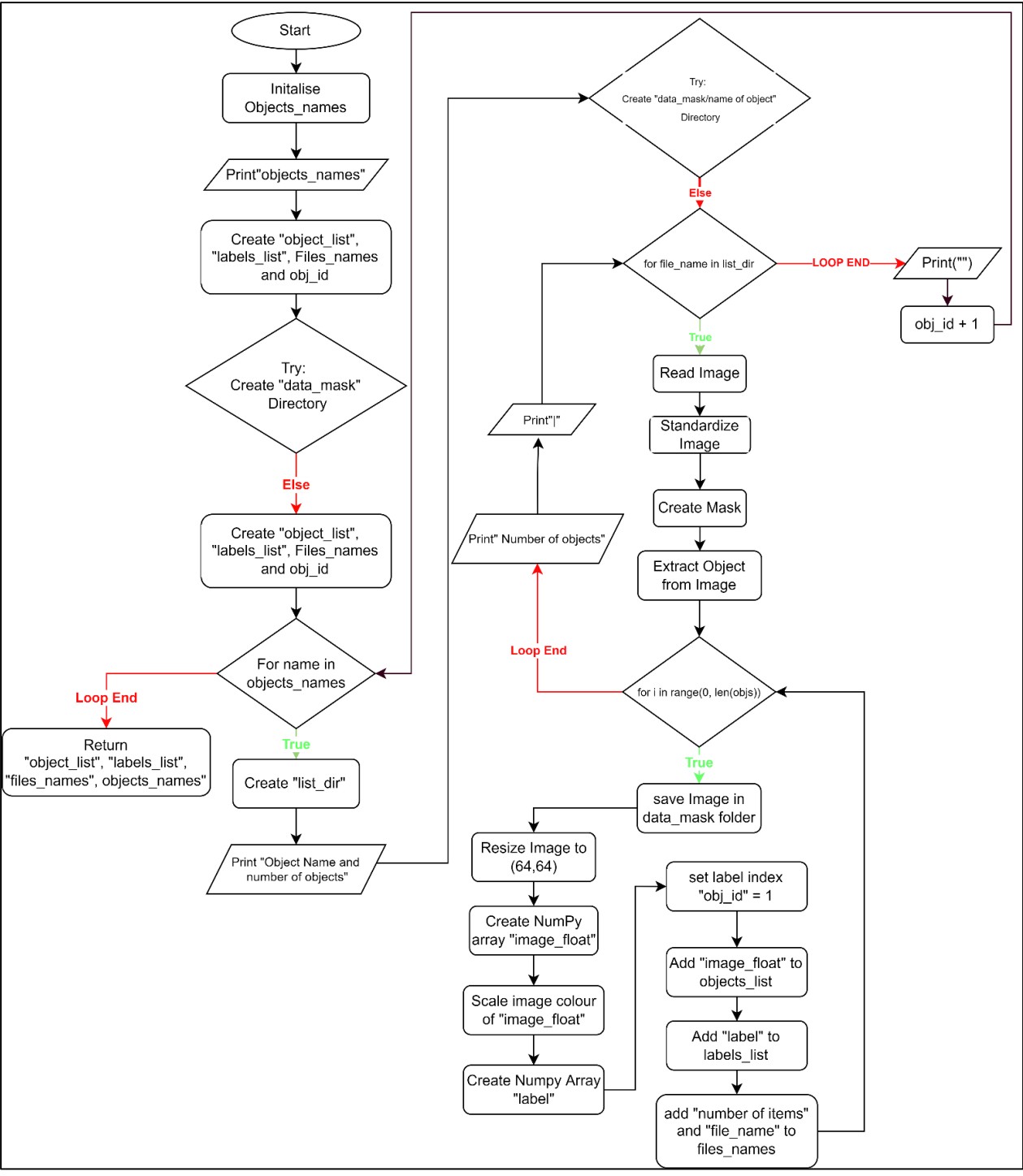

**Figure 3.** Function "load_dataset()" in "MobileNet_Training.py".

Figure 4 shows the flow chart for the shuffle function, which is originally from Niryo One. The shuffle function is used to randomize the dataset; this is used to increase the model's accuracy.

1. Create a Zip the input dataset and convert it to a list. After that pass it to a variable "c".
2. Shuffle the list c.
3. Zip the list c and return the list.

A class called "MyModel" is created. This class is used to create instances of the MobileNet Model. Figure 5 shows functionalities of the class.

1. When the class is called, the arguments for learning rate, Epoch, Batch_size, and Num_objectNames must be filled. The class will then initialize these values as variables under the same name.
2. The bottom layer of MobileNet will first be created ("Bottom_model"). The bottom layer is important as it acts as the convolutional layer. This layer performs feature extraction.
3. The Top Layer of MobileNet is then created using the output of the bottom layer ("Bottom_model").
4. A flatten layer is added to convert the output map from the bottom layer to a one-dimensional array of numbers and vectors.
5. A dense hidden layer of 128 neurons with activation function of "ReLU" is added. The "ReLU" function is added in the hidden layers as its less susceptible to vanishing gradient problems and performs calculations faster compared with other activation functions.
6. A dropout layer is added to prevent overfitting in the model.
7. The final output layer is then added with an activation function of "SoftMax". The "SoftMax" activation function is always used as it converts the output to a normalized probability distribution.
8. The bottom and the top layers are then combined.
9. The bottom layers are frozen as the convolutional layer does not need to be trained.
10. An optimizer variable is created using the Adam function. Adam optimizer is a gradient descent optimizer and its widely used as its more efficient and consumes lesser memory.
11. The MobileNet Model is then compiled.

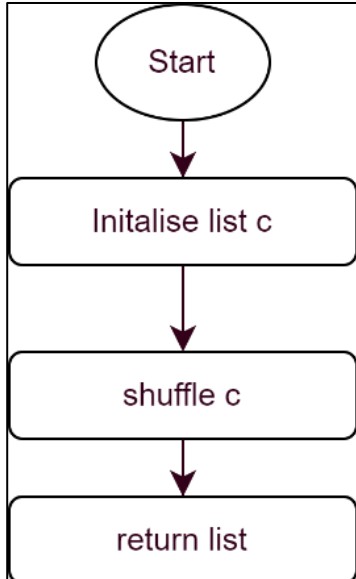

**Figure 4.** Shuffle function in "MobileNet_Training.py".

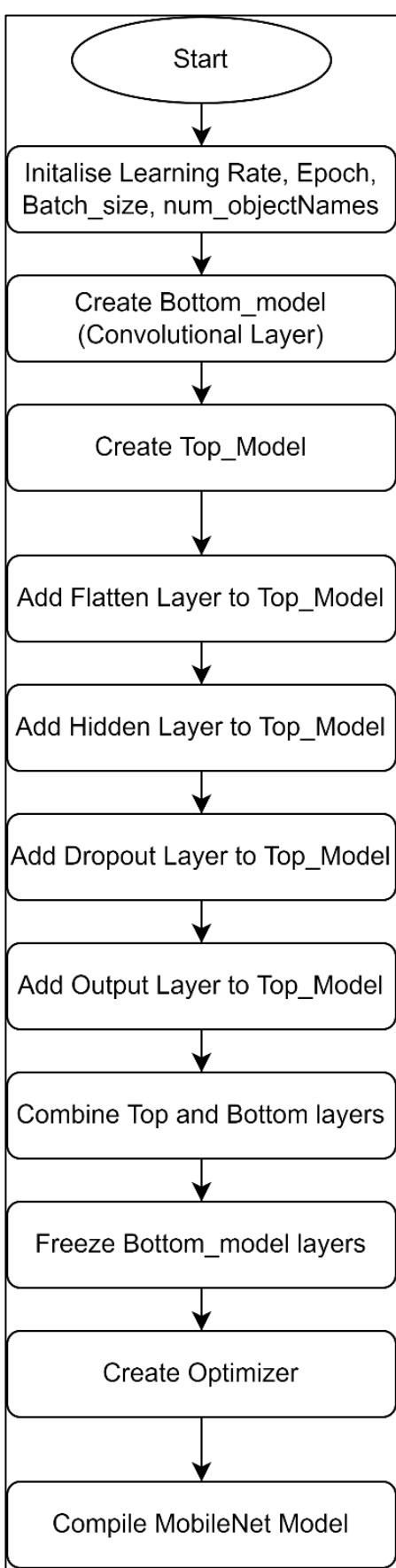

**Figure 5.** Class "MyModel" in "MobileNet_Training.py".

## 4. Results

In this paper, three objects named "IC_Chip", "Circle_Wafer", and "Square_Wafer" are chosen to test the program. Figure 6 shows the "Play" menu in Niryo One's graphical user interface (GUI). As seen, the images on the right are the live feeds of the workspace. If an object is placed on the workspace, it will be detected by the object recognition program. Once the program detects the object, the object's name and prediction percentage will appear in the GUI. When the button for the respective object is pressed in the left-hand-side menu, the robot will pick up this particular object from the workspace and place it on the conveyor belt or the pre-defined position.

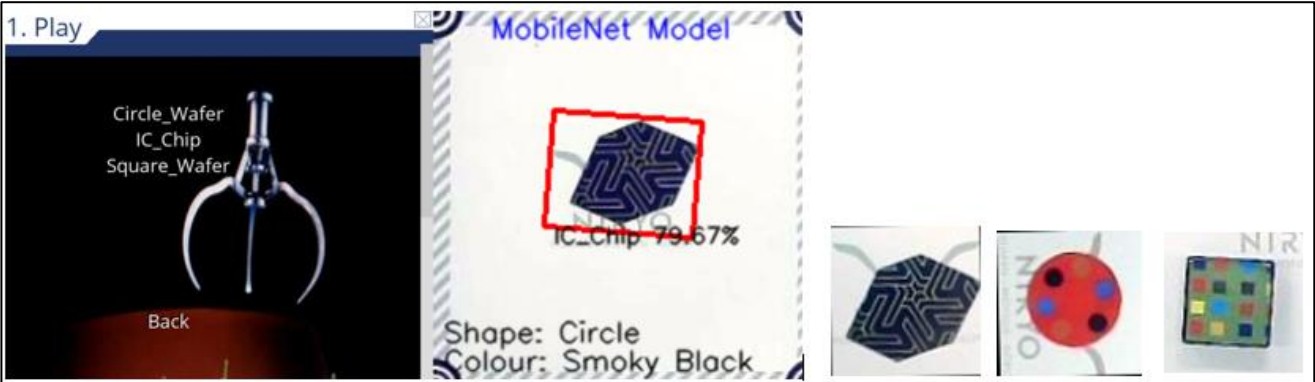

**Figure 6.** "Play" menu in GUI.

Next is the "Settings" menu, as shown in Figure 7. Among the options, "Sequential Training" is for the original machine learning algorithm in Niryo one, and "MobileNet Training" is for the developed algorithm using MobileNet model. When the model is being trained by either of these two models, the dataset will be split into two, one is for training and the other is for testing. Number "0" is used to represent if there is no edge/shape detected in the image, and the associated images will form the training set. Integers "1", "2", etc. show the number of edges/shapes detected in the image, and these images will form the testing set.

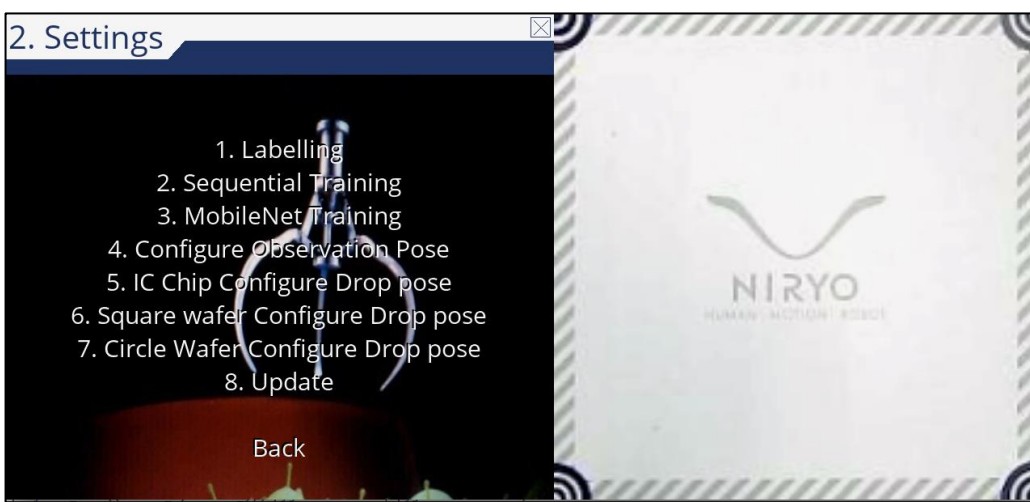

**Figure 7.** "Settings" menu in GUI.

Usually, a good dataset after the training process returns majority of the images as testing data. Table 1 shows the training results for the MobleNet model.

**Table 1.** MobileNet model training result.

| MobileNet Model |
| --- |
| Total Time Taken: 21.0 s |
| 1/1 [==============================]—ls 656 ms/step |
| len 1 3 |
| ['Circle_wafer', "IC_Chip", "Square_Wafer"] |
| Circle_Wafer 32 |
| 1\|1\|1\|1\|1\|1\|1\|1\|1\|1\|1\|1\|1\|1\|1\|0\|1\|1\|1\|1\|1\|2\|1\|1\|2\|2\|1\|1\|1\|1\|1\|1\|1\| |
| IC_Chip 38 |
| 1\|1\|1\|1\|1\|1\|1\|1\|1\|2\|1\|1\|2\|1\|1\|1\|1\|1\|1\|1\|1\|1\|2\|1\|1\|1\|1\|1\|1\|1\|1\|1\|1\|2\|1\|1\|0\|1\|1\|1\|1\| |
| Square_Wafer 38 |
| 1\|1\|1\|1\|1\|1\|1\|1\|1\|1\|1\|1\|1\|1\|1\|1\|1\|1\|1\|1\|1\|1\|1\|1\|1\|1\|1\|1\|1\|1\|1\|1\|1\|1\|1\|1\|1\|1\|1\|1\|1\| |

Figure 8 shows the "Configure Observation Pose" function, which allows the user to switch workspace environment provided if the workspace consists of four debug markers. In this robotic arm system, the user is able to interchange with two workspaces, i.e., conveyor belt and a stationary workspace.

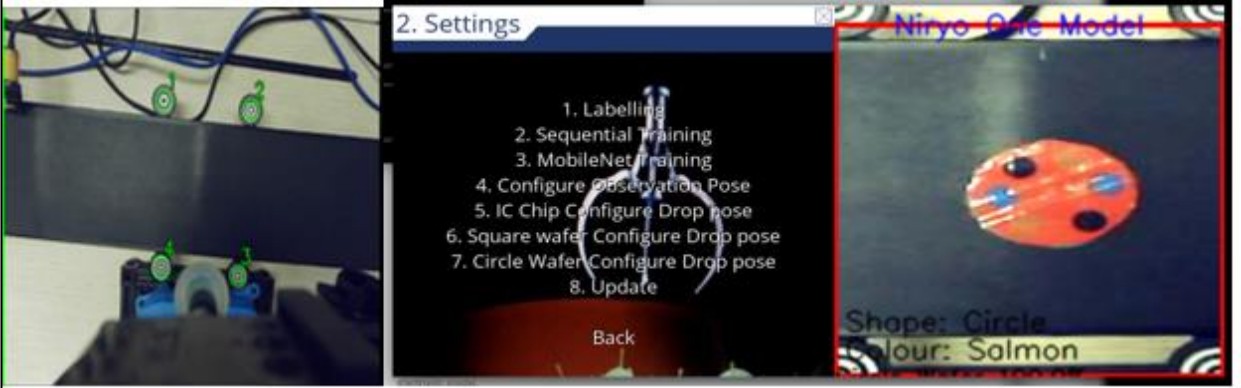

**Figure 8.** Configure observation pose function.

Next is to configure the drop pose for three individual objects. As seen in Figure 9, if the user chooses any object's "Configure Drop pose", the GUI will turn green, and the robot is set in the learning mode. The user is then required to guide the robot arm to the new desired position for the object to be released. If pressing the "enter" key, it overwrites the pre-defined position that was initially coded in the program. However, once the program exits, it reverts to the original pre-defined position.

In this experiment, the learning rate is set as $1 \times 10^{-4}$. The epoch is set as three times that of the total number of objects. The batch size is 32.

Table 2 shows the training performance of the MobileNet model. The confusion matrix shows that the MobileNet model has an accuracy score of 91%. Table 3 shows the training performance of the Niryo One's existing sequential model, which shows an accuracy score of 65%. The MobileNet Model has a 26% improvement in accuracy compared to the sequential model.

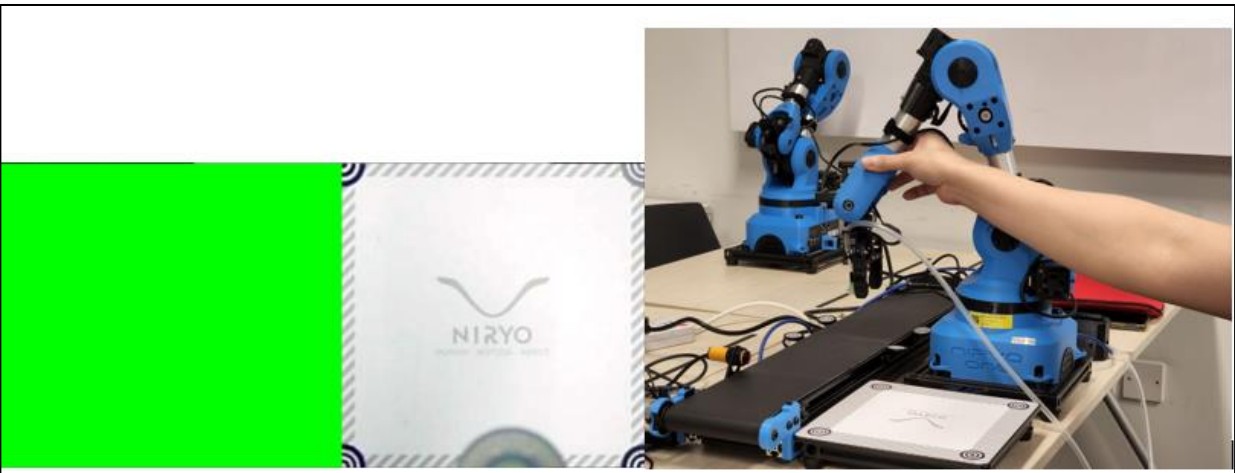

**Figure 9.** Configure object drop pose function.

**Table 2.** MobileNet model training performance.

|  | Precision | Recall | f1-Score | Support |
|---|---|---|---|---|
| Circle_wafer | 1.00 | 0.71 | 0.83 | 7 |
| IC_Chip | 0.80 | 1.00 | 0.89 | 8 |
| Square_wafer | 1.00 | 1.00 | 1.00 | 8 |
| accuracy |  |  | 0.91 | 23 |
| macro avg | 0.93 | 0.90 | 0.91 | 23 |
| weighted avg | 0.93 | 0.91 | 0.91 | 23 |
| MobleNet model |  |  |  |  |
| Total Time Taken: 19.362 s |  |  |  |  |

**Table 3.** Sequential model training performance.

|  | Precision | Recall | f1-Score | Support |
|---|---|---|---|---|
| Circle_wafer | 1.00 | 0.71 | 0.83 | 7 |
| IC_Chip | 0.80 | 1.00 | 0.89 | 8 |
| Square_wafer | 1.00 | 1.00 | 1.00 | 8 |
| accuracy |  |  | 0.91 | 23 |
| macro avg | 0.93 | 0.90 | 0.91 | 23 |
| weighted avg | 0.93 | 0.91 | 0.91 | 23 |
| Niryo One Sequential model |  |  |  |  |
| Total Time Taken: 7.191 s |  |  |  |  |

Figure 10 shows the scatterplot, which compares the accuracy vs. time for 40 observations of the MobileNet and Sequential model. It could be seen from the graph that the Niryo One's sequential model has less accuracy and has a wider spread of accuracy. The MobileNet model, although it takes a longer time, has better accuracy and a more clustered spread of accuracy. This concludes that the MobileNet model has a higher precision and accuracy.

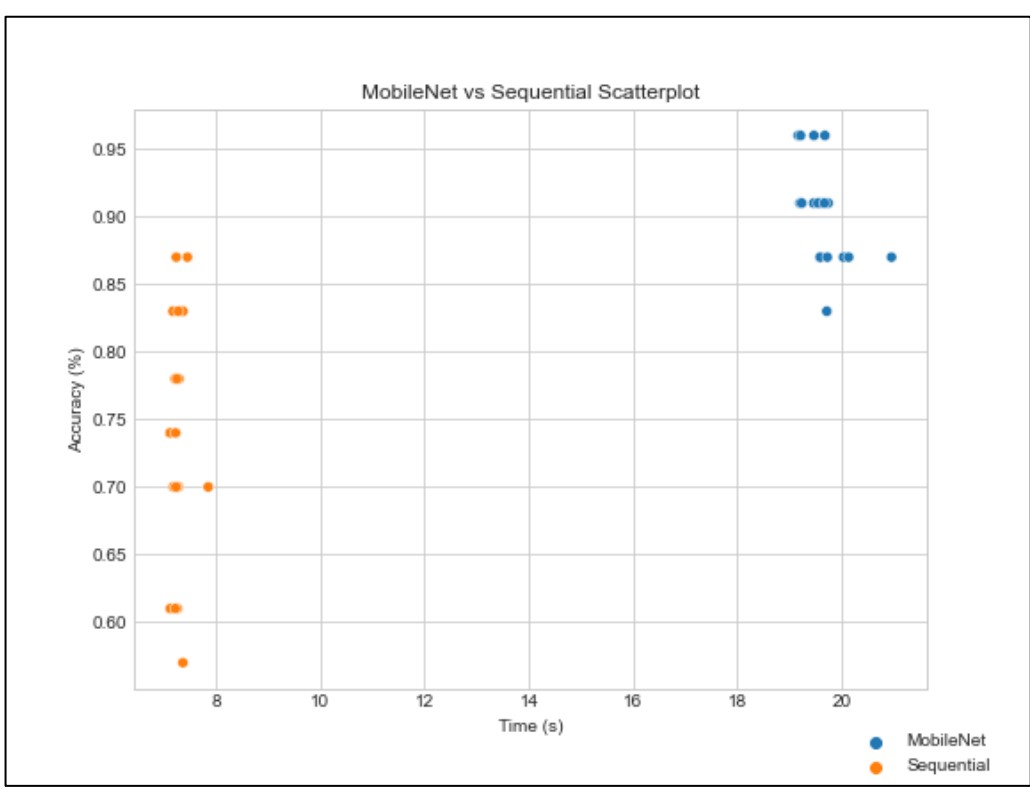

**Figure 10.** Scatterplot of MobileNet vs. Sequential Model.

## 5. Discussion

As shown in the last section, the existing Niryo One robotic arm system can work on any surface as long as it has four debug markers to perform camera calibration. In actual pick-and-place applications, the workspace is subject to change or being altered. Therefore, the constraint of having four markers is preferable to be removed so that the intelligent pick-and-place system is able to work on a broader range of applications.

It was found in the experimental study that sometimes both models failed to detect the objects, or took a long time to detect the objects. We tried to change the lighting condition or shift the location of the objects in order to solve this issue. It was also found that the live feed of the Niryo one's camera was slow, probably due to the low-quality graphics driver in the microprocessor.

As the MobileNet model had to be coded and integrated into the Niryo One's existing platform, there were compatibility issues among the OS, the programming language, and various libraries and platforms. Some libraries or packages were backtracked with older versions so that they could work together with the existing platform. For example, the Niryo One Visual pick-and-place demonstration code does not work in newer versions of Pygame and Pygame-menu library. The ROS version used in Niryo one is the outdated ROS Kinetic. Solutions were found by installing older versions of the libraries and using Anaconda to create a virtual environment with Python 3.6. We also wish to compare MobileNet model with other neural-network based training models in the future and to integrate it into commercial robots.

Another compatibility issue lay in the Python and OpenCV in the GUI file. It was resolved by editing the file and importing it to cv2 (i.e., the module import name for opencv-python).

In addition, there was an overfitting issue in the training process, resulting in inaccurate predictions of the model. To fix this, we tried to collect good data by taking around 300 images of the object and identifying the data masks of the object. The process only re-

turned 5 to 10 mask images. These good masks were picked up with their names identified in the data folder and copied into the folder to store good data.

## 6. Conclusions

This paper has developed an intelligent object detection and picking system based on machine learning. The MobileNet model for training purposes has been built with a properly chosen number of neurons and activation functions. The training program has been coded in Python in the TensorFlow platform and has been implemented and integrated into an educational six-axis robotic arm. An experimental test has been conducted with three different objects to train the model and identify the objects. The testing results have shown that the MobileNet model outperformed the robot's existing sequential model, as can be seen from the accuracy score in the confusion matrix. The statistical study has shown that the MobileNet achieved a higher precision with a more clustered spread of accuracy.

**Author Contributions:** Writing—original draft, F.H.; Writing—review & editing, D.W.L.T. and A.A. All authors have read and agreed to the published version of the manuscript.

**Funding:** This research received no external funding.

**Data Availability Statement:** The data generated from this study are available from the corresponding author on request.

**Conflicts of Interest:** The authors declare no conflict of interest.

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
