# Peer review of "Intelligent Pick-and-Place System Using MobileNet"

_electronics, doi:10.3390/electronics12030621_

Round 1

Reviewer 1 Report

Dear Authors,

Please find the attached file for your reference.

Regards

Author Response

The authors would like to express their sincere appreciation to the reviewer for his/her critical comments and the efforts provide in helping us to improve the quality and presentation of the paper further.

Please see our detailed responses in the attached word document.

Reviewer 2 Report

The authors describe a neural-net-based pick and place system. Advantages of the article:1) A novel neural-net-based object detection system. 2) Implementation of the system. 3) Comparison of the system with the original Niryo One. Disadvantages of the text: 1) The authors need to compare the system with other neural-net-based systems for robotic-arm control known from the literature. 2) The reviewer does not know the precise meaning of Figure 7, especially what means “time” on the x-axis. 3) The authors do not mention the learning database in the text.

Author Response

(The authors gave the same response as above.)

Reviewer 3 Report

Title: Intelligent Pick & Place System using MobileNet 2

The topic of the paper is important from both theoretical and practical perspectives. The manuscript could be improved by considering the following points:

1. In the abstract you may state the novelty of the work. The abstract needs to be corrected with the required information. Less important lines can be removed. Make the abstract and proposal relevant.

2. Problem statement should be discussed in the first para of the Introduction part. Include the main objects of the work.

3. Related work must discuss the existing methods with their advantages and disadvantages. You can modulate the one para about existing limitations and proposed ideology

4. Architecture model (figure 1) is not clearly visualised and understandable. you could consider including model with good resolution. More explanation and discussion can be presented in this section.

5. Some equation terms must be ensured and defined before using the equation.

6. Dataset details should be discussed in the result section and what are the parameters used for experimental evaluation should discuss.

7. There are no advantages and a disadvantage discussed in the result section. There is no comparative analysis. Kindly include all the parts in the result analysis.

8. The quality of figures is less than normal, increasing the quality of figures.

9. Future scope of the article can be discussed with limitations in the conclusion part.

10. In some references, year is written at the end and in some references in between. Please follow uniform format.

11. This paper needs rigorous revision in terms of techniques, English and presentation.

Author Response

(The authors gave the same response as above.)

Round 2

Reviewer 1 Report

Dear Authors,

Thank you for addressing all my comments and I don't have any further concerns about your paper.

Regards  

Author Response

The authors would like to express their sincere appreciation to the reviewer for his/her recognition: “Thank you for addressing all my comments”. Your critical comments have helping us greatly to improve the quality and presentation of the paper.

Reviewer 2 Report

The authors addressed all my doubts.

Author Response

The authors would like to express their sincere appreciation to the reviewer for his/her recognition: “The authors addressed all my doubts.” Your critical comments have helping us greatly to improve the quality and presentation of the paper.

We have checked through our paper to clear all the format/language/spelling errors as much as we can.   

Reviewer 3 Report

The article is well written, but it needs minor revision. 1- Pay attention to English, bad construction of many sentences. 2- Figures should be enhanced or presented in another way. 3- Section 5 looks a pale part, the drawing objects should be improved and presented in better clarification 4- Recheck the format of writing equations and presentation 5- Revise your paper to improve organization and cohesion 6- Too many punctuation and grammar in the manuscript. Rectify them.

Author Response

The authors would like to express their sincere appreciation to the reviewer for his/her recognition: “The article is well written.”, as well as his/her critical comments, which have helping us greatly to improve the quality and presentation of the paper.

The paper has now been revised again with some minor changes, especially in format / presentation / language / spelling, to the best of our ability.

Please find our detailed responses to the individual comments in the attached document.
